# Third-order Smoothness Helps: Faster Stochastic Optimization Algorithms for Finding Local Minima

**Yaodong Yu**\*
Department of Computer Science
University of Virginia
Charlottesville, VA 22904
yy8ms@virginia.edu

**Pan Xu**\*
Department of Computer Science
University of California, Los Angeles
Los Angeles, CA 90095
panxu@cs.ucla.edu

**Quanquan Gu**
Department of Computer Science
University of California, Los Angeles
Los Angeles, CA 90095
qgu@cs.ucla.edu

## Abstract

We propose stochastic optimization algorithms that can find local minima faster than existing algorithms for nonconvex optimization problems, by exploiting the third-order smoothness to escape non-degenerate saddle points more efficiently. More specifically, the proposed algorithm only needs $\widetilde{O}(\epsilon^{-10/3})$ stochastic gradient evaluations to converge to an approximate local minimum $\mathbf{x}$, which satisfies $\|\nabla f(\mathbf{x})\|_2 \leq \epsilon$ and $\lambda_{\min}(\nabla^2 f(\mathbf{x})) \geq -\sqrt{\epsilon}$ in unconstrained stochastic optimization, where $\widetilde{O}(\cdot)$ hides logarithm polynomial terms and constants. This improves upon the $\widetilde{O}(\epsilon^{-7/2})$ gradient complexity achieved by the state-of-the-art stochastic local minima finding algorithms by a factor of $\widetilde{O}(\epsilon^{-1/6})$. Experiments on two nonconvex optimization problems demonstrate the effectiveness of our algorithm and corroborate our theory.

## 1  Introduction

We study the following unconstrained stochastic optimization problem

$$\min_{\mathbf{x} \in \mathbb{R}^d} f(\mathbf{x}) = \mathbb{E}_{\xi \sim \mathcal{D}}[F(\mathbf{x}; \xi)], \tag{1.1}$$

where $F(\mathbf{x}; \xi) : \mathbb{R}^d \to \mathbb{R}$ is a stochastic function and $\xi$ is a random variable sampled from a fixed distribution $\mathcal{D}$. In particular, we are interested in nonconvex optimization where the expected function $f(\mathbf{x})$ is not convex. This kind of nonconvex optimization is ubiquitous in machine learning, especially deep learning [24]. Finding a global minimum of nonconvex problem (1.1) is generally NP hard [18]. Nevertheless, for many nonconvex optimization problems in machine learning, a local minimum is adequate and can be as good as a global minimum in terms of generalization performance, such as in deep learning [10, 13].

In this paper, we aim to design efficient stochastic optimization algorithms that can find an approximate local minimum of (1.1), i.e., an $(\epsilon, \epsilon_H)$-second-order stationary point $\mathbf{x}$ defined as follows

$$\|\nabla f(\mathbf{x})\|_2 \leq \epsilon, \text{ and } \lambda_{\min}(\nabla^2 f(\mathbf{x})) \geq -\epsilon_H, \tag{1.2}$$

---

where $\epsilon, \epsilon_H \in (0, 1)$. Notably, when $\epsilon_H = \sqrt{L_2\epsilon}$ for Hessian Lipschitz $f$ with parameter $L_2$, (1.2) is equivalent to the definition of $\epsilon$-second-order stationary point [28]. Algorithms based on cubic regularized Newton's method [28] and its variants [1, 7, 12, 23, 33, 31] have been proposed to find such approximate local minima. However, all of them need to solve the cubic problems exactly [28] or approximately [1, 7] in each iteration, which poses a rather heavy computational overhead. Another line of research employs the negative curvature direction to find the local minimum by combining accelerated gradient descent and negative curvature descent [8, 2], which yet becomes impractical in large scale and high dimensional machine learning problems due to the frequent computation of negative curvature in each iteration.

To alleviate the computational burden of local minimum finding algorithms, there has emerged a fresh line of research [34, 5, 21] that tries to achieve the iteration complexity as the state-of-the-art second-order methods, while only utilizing first-order oracles. The key observation is that first-order methods with noise injection [15, 20] are essentially an equivalent way to extract the negative curvature direction around saddle points [34, 5]. Together with the Stochastically Controlled Stochastic Gradient (SCSG) method [25], the aforementioned methods [34, 5] converge to an $(\epsilon, \sqrt{\epsilon})$-second-order stationary point (an approximate local minimum) within $\widetilde{O}(\epsilon^{-7/2})$ stochastic gradient evaluations, where $\widetilde{O}(\cdot)$ hides logarithm polynomial factors and constants. In this work, motivated by [9] which employed the third-order smoothness of $f$ in deterministic nonconvex optimization to find a first-order stationary point, we explore the benefits of third-order smoothness in finding an approximate local minimum in the stochastic nonconvex optimization. In particular, we propose a stochastic optimization algorithm, named as **FLASH**, which only utilizes first-order oracles and finds the $(\epsilon, \epsilon_H)$-second-order stationary point within $\widetilde{O}(\epsilon^{-10/3})$ stochastic gradient evaluations. Note that our gradient complexity matches that of the state-of-the-art stochastic optimization algorithm SCSG [25] for finding first-order stationary points. At the core of our algorithm is an exploitation of the third-order smoothness of the objective function $f$ which enables us to choose a larger step size in the negative curvature descent stage, and therefore leads to a faster convergence rate. The main contributions of our work are as follows

- We show that the third-order smoothness of the nonconvex function can lead to a faster escape from saddle points in the stochastic optimization. We characterize, for the first time, the improvement brought by third-order smoothness in finding the approximate local minimum.

- We propose an efficient stochastic algorithm for general stochastic objective functions and prove faster convergence rates for finding local minima. More specifically, for stochastic optimization, our algorithm converges to an approximate local minimum with only $\widetilde{O}(\epsilon^{-10/3})$ stochastic gradient evaluations.

- In each outer iteration, our proposed algorithm only performs either one step of negative curvature descent, or an epoch of SCSG, which saves a lot of gradient and negative curvature computations compared with existing algorithms.

**Notation** For a vector $\mathbf{x} = (x_1, ..., x_d)^\top \in \mathbb{R}^d$, we denote the $\ell_q$ norm as $\|\mathbf{x}\|_q = (\sum_{i=1}^d |x_i|^q)^{1/q}$ for $0 < q < +\infty$. For a matrix $\mathbf{A} = [A_{ij}] \in \mathbb{R}^{d \times d}$, we use $\|\mathbf{A}\|_2$ and $\|\mathbf{A}\|_F$ to denote the spectral and Frobenius norm. For a three-way tensor $\mathcal{T} \in \mathbb{R}^{d \times d \times d}$ and vector $\mathbf{x} \in \mathbb{R}^d$, we denote their inner product as $\langle \mathcal{T}, \mathbf{x}^{\otimes 3} \rangle$. For a symmetric matrix $\mathbf{A}$, let $\lambda_{\max}(\mathbf{A})$ and $\lambda_{\min}(\mathbf{A})$ be the maximum, minimum eigenvalues of matrix $\mathbf{A}$. We use $\mathbf{A} \succeq 0$ to denote $\mathbf{A}$ is positive semidefinite. For two sequences $\{a_n\}$ and $\{b_n\}$, we denote $a_n = O(b_n)$ if $a_n \leq C b_n$ for some constant $C$ independent of $n$. The notation $\widetilde{O}(\cdot)$ hides logarithmic factors. Additionally, we denote $a_n \lesssim b_n$ ($a_n \gtrsim b_n$) if $a_n$ is less than (larger than) $b_n$ up to a constant.

## 2 Related Work

In this section, we discuss related work for finding approximate second-order stationary points in nonconvex optimization. In general, existing literature can be divided into the following three categories.

**Hessian-based:** The pioneer work of [28] proposed the cubic regularized Newton's method to find an $(\epsilon, \epsilon_H)$-second-order stationary point in $O\big(\max\{\epsilon^{-3/2}, \epsilon_H^{-3}\}\big)$ iterations. Curtis et al. [12] showed that the trust-region Newton method can achieve the same iteration complexity as the cubic

regularization method. Recently, Kohler and Lucchi [23], Xu et al. [33] showed that by using subsampled Hessian matrix instead of the entire Hessian matrix in cubic regularization method and trust-region method, the iteration complexity can still match the original exact methods under certain conditions. Zhou et al. [36] improved the second-order oracle complexity (including gradient and Hessian evaluations) by proposing a variance-reduced Cubic regularization method. However, these methods need to compute the Hessian matrix and solve a very expensive subproblem either exactly or approximately in each iteration, which can be computationally intractable for high-dimensional problems.

**Hessian-vector product-based:** Through different approaches, Carmon et al. [8] and Agarwal et al. [1] independently proposed algorithms that are able to find $(\epsilon, \sqrt{\epsilon})$-second-order stationary points within $\widetilde{O}(\epsilon^{-7/4})$ full gradient and Hessian-vector product evaluations. By making an additional assumption of the third-order smoothness on the objective function and combining the negative curvature descent with the "convex until proven guilty" algorithm, Carmon et al. [9] proposed an algorithm that is able to find an $(\epsilon, \sqrt{\epsilon})$-second-order stationary point within $\widetilde{O}(\epsilon^{-5/3})$ full gradient and Hessian-vector product evaluations.[2] For nonconvex finite-sum optimization problems, Agarwal et al. [1] proposed an algorithm which is able to find approximate local minima within $\widetilde{O}(n\epsilon^{-3/2} + n^{3/4}\epsilon^{-7/4})$ stochastic gradient and stochastic Hessian-vector product evaluations, where $n$ is the number of component functions. Reddi et al. [30] proposed an algorithm, which combines first-order and second-order methods to find approximate $(\epsilon, \epsilon_H)$-second-order stationary points, and requires $\widetilde{O}(n^{2/3}\epsilon^{-2} + n\epsilon_H^{-3} + n^{3/4}\epsilon_H^{-7/2})$ stochastic gradient and stochastic Hessian-vector product evaluations. In the general stochastic optimization setting, Allen-Zhu [2] proposed an algorithm named Natasha2, which is based on variance reduction and negative curvature descent, and is able to find $(\epsilon, \sqrt{\epsilon})$-second-order stationary points with at most $\widetilde{O}(\epsilon^{-7/2})$ stochastic gradient and stochastic Hessian-vector product evaluations. Tripuraneni et al. [31] proposed a stochastic cubic regularization algorithm to find $(\epsilon, \sqrt{\epsilon})$-second-order stationary points and achieved the same runtime complexity as [2].

**Gradient-based:** For general nonconvex problems, Ghadimi and Lan [16] proposed a randomized stochastic gradient method and established the complexity of this method for finding a first-order stationary point. Levy [26], Jin et al. [20, 21] showed that it is possible to escape from saddle points and find local minima only using gradient evaluations plus random perturbation. The best-known runtime complexity of these methods is $\widetilde{O}(\epsilon^{-7/4})$ when $\epsilon_H = \sqrt{\epsilon}$ [21]. For nonconvex finite-sum problems, Allen-Zhu and Li [5] proposed a first-order negative curvature finding method called Neon2 and combined it with the stochastic variance reduced gradient (SVRG) method [22, 29, 3, 25], leading to an algorithm that finds $(\epsilon, \epsilon_H)$-second-order stationary points within $\widetilde{O}(n^{2/3}\epsilon^{-2} + n\epsilon_H^{-3} + n^{3/4}\epsilon_H^{-7/2} + n^{5/12}\epsilon^{-2}\epsilon_H^{-1/2})$ stochastic gradient evaluations. For nonconvex stochastic optimization problems, a variant of stochastic gradient descent (SGD) [15] is proved to find the $(\epsilon, \sqrt{\epsilon})$-second-order stationary point within $O(\epsilon^{-4}\text{poly}(d))$ stochastic gradient evaluations. More recently, Xu and Yang [34], Allen-Zhu and Li [5] turned the first-order stationary point finding method SCSG [25] into approximate local minima finding algorithms, which only involves stochastic gradient computation. The runtime complexity of these algorithms is $\widetilde{O}(\epsilon^{-10/3} + \epsilon^{-2}\epsilon_H^{-3})$. In order to further save gradient and negative curvature computations, [35] considered the number of saddle points encountered in the algorithm and proposed the gradient descent with one-step escaping algorithm (GOSE) that saves negative curvature computation. However, none of the above algorithms explore the third-order smoothness of the nonconvex objective function.

## 3  Preliminaries

In this section, we present definitions that will be used in our algorithm design and later theoretical analysis.

**Definition 3.1** (Smoothness). A differentiable function $f$ is $L_1$-smooth, if for any $\mathbf{x}, \mathbf{y} \in \mathbb{R}^d$:

$$\|\nabla f(\mathbf{x}) - \nabla f(\mathbf{y})\|_2 \le L_1 \|\mathbf{x} - \mathbf{y}\|_2.$$

**Definition 3.2** (Hessian Lipschitz). A twice-differentiable function $f$ is $L_2$-Hessian Lipschitz, if for any $\mathbf{x}, \mathbf{y} \in \mathbb{R}^d$:

$$\|\nabla^2 f(\mathbf{x}) - \nabla^2 f(\mathbf{y})\|_2 \leq L_2 \|\mathbf{x} - \mathbf{y}\|_2.$$

Note that Hessian-Lipschitz is also referred to as the second-order smoothness. The above two smoothness conditions are widely used in nonconvex optimization problems [28]. In this paper, we will further explore the effectiveness of third-order derivative Lipschitz condition in nonconvex optimization. We use a three-way tensor $\nabla^3 f(\mathbf{x}) \in \mathbb{R}^{d \times d \times d}$ to denote the third-order derivative of a function, which is formally defined below.

**Definition 3.3** (Third-order Derivative). The third-order derivative of function $f \colon \mathbb{R}^d \to \mathbb{R}$ is a three-way tensor $\nabla^3 f(\mathbf{x}) \in \mathbb{R}^{d \times d \times d}$ which is defined as

$$[\nabla^3 f(\mathbf{x})]_{ijk} = \frac{\partial}{\partial x_i \partial x_j \partial x_k} f(\mathbf{x}),$$

for $i, j, k = 1, \ldots, d$ and $\mathbf{x} \in \mathbb{R}^d$.

Next we introduce the definition of third-order smoothness for function $f$, which implies that the third-order derivative will not change rapidly.

**Definition 3.4** (Third-order Derivative Lipschitz). A thrice-differentiable function $f$ has $L_3$-Lipschitz third-order derivative, if for any $\mathbf{x}, \mathbf{y} \in \mathbb{R}^d$:

$$\|\nabla^3 f(\mathbf{x}) - \nabla^3 f(\mathbf{y})\|_F \leq L_3 \|\mathbf{x} - \mathbf{y}\|_2.$$

The above definition has been introduced in [6], and the third-order derivative Lipschitz is also referred to as third-order smoothness in [9]. One can also use another equivalent notion of third-order derivative Lipschitz condition used in [9]. Note that the third-order Lipschitz condition is critical in our algorithms and theoretical analysis in later sections. In the sequel, we will use third-order derivative Lipschitz and third-order smoothness interchangeably.

**Definition 3.5** (Optimal Gap). For a function $f$, we define the optimal gap $\Delta_f$ at point $\mathbf{x}_0$ as

$$f(\mathbf{x}_0) - \inf_{\mathbf{x} \in \mathbb{R}^d} f(\mathbf{x}) \leq \Delta_f.$$

Without loss of generality, we assume $\Delta_f < +\infty$.

**Definition 3.6** (Geometric Distribution). For a random integer $X$, define $X$ has a geometric distribution with parameter $p$, denoted as $\mathrm{Geom}(p)$, if it satisfies that

$$\mathbb{P}(X = k) = p^k (1 - p), \quad \forall k = 0, 1, \ldots.$$

**Definition 3.7** (Sub-Gaussian Stochastic Gradient). For any $\mathbf{x} \in \mathbb{R}^d$ and random variable $\xi \in \mathcal{D}$, the stochastic gradient $\nabla F(\mathbf{x}; \xi)$ is sub-Gaussian with parameter $\sigma$ if it satisfies

$$\mathbb{E}\left[ \exp\left( \frac{\|\nabla F(\mathbf{x}; \xi) - \nabla f(\mathbf{x})\|_2^2}{\sigma^2} \right) \right] \leq \exp(1).$$

In addition, we introduce $\mathbb{T}_g$ to denote the time complexity of stochastic function value and gradient evaluation, i.e., $(F(\mathbf{x}; \xi_i), \nabla F(\mathbf{x}; \xi_i))$ for $\xi_i \in \mathcal{D}$, and $\mathbb{T}_h$ to denote the time complexity of stochastic Hessian-vector product evaluation, i.e., $\nabla^2 F(\mathbf{x}; \xi_i)\mathbf{v}$ for a given vector $\mathbf{v}$ and $\xi_i \in \mathcal{D}$.

## 4 Exploiting Third-order Smoothness

In this section we will show how to employ the third-order smoothness of the objective function to make better use of the negative curvature direction for escaping saddle points. We first give an enlightening explanation on why third-order smoothness helps in general nonconvex optimization problems. Then we present our main algorithm which is able to utilize the third-order smoothness to take a larger step size for general stochastic optimization.

In order to find local minima in nonconvex problems, different kinds of approaches have been explored to escape from saddle points. One of these approaches is to use negative curvature direction [27] to escape from saddle points, which has been explored in many existing studies [8, 11, 2].

According to recent work by [34, 5], one can extract the negative curvature direction by only using stochastic gradient evaluations, which makes the negative curvature descent approach more appealing.

We first consider a simple case to illustrate how to utilize the third-order smoothness when taking a negative curvature descent step. For nonconvex optimization problems, an $\epsilon$-first-order stationary point $\widehat{\mathbf{x}}$ can be found by using first-order methods such as gradient descent. If $\widehat{\mathbf{x}}$ is not an $(\epsilon, \epsilon_H)$-second-order stationary point defined in (1.2), then there must exist a unit vector $\widehat{\mathbf{v}}$ such that

$$\widehat{\mathbf{v}}^\top \nabla^2 f(\widehat{\mathbf{x}}) \, \widehat{\mathbf{v}} \le -\frac{\epsilon_H}{2}.$$

As studied in [8, 34, 5], one can take a negative curvature descent step along the direction of $\widehat{\mathbf{v}}$ to escape from the saddle point $\widehat{\mathbf{x}}$, i.e.,

$$\widetilde{\mathbf{y}} = \underset{\mathbf{y} \in \{\mathbf{u}, \mathbf{w}\}}{\arg\min} f(\mathbf{y}), \ \mathbf{u} = \widehat{\mathbf{x}} - \widetilde{\alpha}\,\widehat{\mathbf{v}}, \ \mathbf{w} = \widehat{\mathbf{x}} + \widetilde{\alpha}\,\widehat{\mathbf{v}}, \tag{4.1}$$

where $\widetilde{\alpha}$ is the step size. Suppose the function $f$ is $L_1$-smooth and $L_2$-Hessian Lipschitz, then the step size can be set as $\widetilde{\alpha} = O(\epsilon_H/L_2)$ and the negative curvature descent step (4.1) is guaranteed to attain the following function value decrease,

$$f(\widetilde{\mathbf{y}}) - f(\widehat{\mathbf{x}}) = -O\left(\frac{\epsilon_H^3}{L_2^2}\right). \tag{4.2}$$

Inspired by the previous work [9], we aim to achieve more function value decrease than (4.2) by incorporating an additional assumption that the objective function has $L_3$-Lipschitz third-order derivatives (third-order smoothness). More specifically, we adjust the negative curvature descent step in (4.1) as follows,

$$\widehat{\mathbf{y}} = \underset{\mathbf{y} \in \{\mathbf{u}, \mathbf{w}\}}{\arg\min} f(\mathbf{y}), \ \mathbf{u} = \widehat{\mathbf{x}} - \alpha\,\widehat{\mathbf{v}}, \ \mathbf{w} = \widehat{\mathbf{x}} + \alpha\,\widehat{\mathbf{v}}, \tag{4.3}$$

where $\alpha = O(\sqrt{\epsilon_H/L_3})$ is the adjusted step size which can be much larger than the step size $\widetilde{\alpha}$ in (4.1) when $\epsilon_H$ is sufficiently small. The adjusted negative curvature descent step (4.3) is guaranteed to decrease the function value by a larger decrement, i.e.,

$$f(\widehat{\mathbf{y}}) - f(\widehat{\mathbf{x}}) = -O\left(\frac{\epsilon_H^2}{L_3}\right). \tag{4.4}$$

Compared with (4.2), the decrement in (4.4) can be substantially larger. In other words, if we make the additional assumption of the third-order smoothness, the negative curvature descent with larger step size will make more progress toward decreasing the function value. Note that [9] focuses on deterministic optimization, while our work is focused on the stochastic optimization. Here we need to carefully design our algorithm to improve the computational complexity in the stochastic setting.

In the following, we will present an algorithm for stochastic nonconvex optimization which exploits the benefits of third-order smoothness to escape from saddle points . Recall the general stochastic optimization problem in (1.1). In this setting, one cannot have access to the full gradient or Hessian information. Instead, only stochastic gradient and stochastic Hessian-vector product evaluations are accessible. As a result, we have to employ stochastic optimization methods to calculate the negative curvature direction. There exist two kinds of methods to calculate the negative curvature direction $\widehat{\mathbf{v}}$ for the general stochastic problem. The first kind is an online PCA method, i.e., Oja's algorithm [4], which uses Hessian-vector product evaluations and can be seen as a stochastic variant of FastPCA [14]. Another method is the online version of the Neon algorithm, denote as Neon2$^{\text{online}}$ [5], which only requires stochastic gradient evaluations.

By using either Oja's algorithm or Neon2$^{\text{online}}$, there exists an algorithm, denoted by ApproxNC-Stochastic, which uses stochastic gradient evaluations or stochastic Hessian-vector product evaluations to find the negative curvature direction for general stochastic nonconvex optimization problem (1.1). Specifically, ApproxNC-Stochastic returns a unit vector $\widehat{\mathbf{v}}$ that satisfies $\widehat{\mathbf{v}}^\top \nabla^2 f(\mathbf{x}) \, \widehat{\mathbf{v}} \le -\epsilon_H/2$ provided $\lambda_{\min}(\nabla^2 f(\mathbf{x})) < -\epsilon_H$, otherwise it will return $\widehat{\mathbf{v}} = \perp$. Based on ApproxNC-Stochastic, we present our negative curvature descent algorithm in Algorithm 1.

Note that the Rademacher random variable $\zeta$ is an important feature in Algorithm 1. As we cannot access the full objective function value in stochastic setting, we use a Rademacher variable ($\zeta = -1$ or $\zeta = 1$ with probability $1/2$) in our algorithm to decide the direction of negative curvature descent step.

**Algorithm 1** NCD3-Stochastic $(f, \mathbf{x}, \{L_i\}_{i=1}^3, \delta, \epsilon_H)$

1: Set $\alpha = \sqrt{3\epsilon_H/L_3}$
2: $\widehat{\mathbf{v}} \leftarrow$ ApproxNC-Stochastic$(f, \mathbf{x}, L_1, L_2, \delta, \epsilon_H)$
3: **if** $\widehat{\mathbf{v}} \neq \perp$
4:     generate a Rademacher random variable $\zeta$
5:     $\widehat{\mathbf{y}} \leftarrow \mathbf{x} + \zeta \alpha \widehat{\mathbf{v}}$
6:     **return** $\widehat{\mathbf{y}}$
7: **else**
8:     **return** $\perp$

Therefore, with the step size $\alpha = O(\sqrt{\epsilon_H/L_3})$ for the negative curvature descent step, Algorithm 1 can make greater progress in expectation when $\lambda_{\min}(\nabla^2 f(\mathbf{x})) < -\epsilon_H$, and we summarize this property as follows.

**Lemma 4.1.** Let $f(\mathbf{x}) = \mathbb{E}_{\xi \sim \mathcal{D}}[F(\mathbf{x}; \xi)]$ and each stochastic function $F(\mathbf{x}; \xi)$ is $L_1$-smooth, $L_2$-Hessian Lipschitz continuous, and the third derivative of $f(\mathbf{x})$ is $L_3$-Lipschitz. Set $\epsilon_H \in (0, 1)$ and step size as $\alpha = \sqrt{3\epsilon_H/L_3}$. If the input $\mathbf{x}$ of Algorithm 1 satisfies $\lambda_{\min}(\nabla^2 f(\mathbf{x})) < -\epsilon_H$, then with probability $1 - \delta$, Algorithm 1 will return $\widehat{\mathbf{y}}$ such that $\mathbb{E}_\zeta[f(\mathbf{x}) - f(\widehat{\mathbf{y}})] \geq 3\epsilon_H^2/8L_3$, where $\delta \in (0, 1)$ and $\mathbb{E}_\zeta$ denotes the expectation over the Rademacher random variable $\zeta$. Furthermore, if we choose $\delta \leq \epsilon_H/(3\epsilon_H + 8L_2)$, it holds that

$$\mathbb{E}[f(\widehat{\mathbf{y}}) - f(\mathbf{x})] \leq -\frac{\epsilon_H^2}{8L_3},$$

where $\mathbb{E}$ is over all randomness of the algorithm, and the total runtime is $\widetilde{O}\big((L_1^2/\epsilon_H^2)\mathbb{T}_h\big)$ if ApproxNC-Stochastic adopts online Oja's algorithm, and $\widetilde{O}\big((L_1^2/\epsilon_H^2)\mathbb{T}_g\big)$ if ApproxNC-Stochastic adopts Neon2$^{\text{online}}$.

## 5 Fast Local Minima Finding Algorithm

In this section, we present our main algorithm to find approximate local minima for nonconvex stochastic optimization problems, based on the negative curvature descent algorithms proposed in previous section.

To find the local minimum, we use SCSG [25], which is the state-of-the-art stochastic optimization algorithm, to find a first-order stationary point and then apply Algorithm 1 to escape the saddle point using negative curvature direction. The proposed method is presented in Algorithm 2, We use a subsampled stochastic gradient $\nabla f_{\mathcal{S}}(\mathbf{x})$ in the outer loop (Line 4) of Algorithm 2, which is defined as $\nabla f_{\mathcal{S}}(\mathbf{x}) = 1/|\mathcal{S}| \sum_{i \in \mathcal{S}} \nabla F(\mathbf{x}; \xi_i)$.

As shown in Algorithm 2, we use subsampled gradient to check whether $\mathbf{x}_{k-1}$ is a first-order stationary point. Suppose the stochastic gradient $\nabla F(\mathbf{x}; \xi)$ satisfies the gradient sub-Gaussian condition (3.7) and the batch size $|\mathcal{S}_k|$ is large enough, then $\|\nabla f(\mathbf{x}_{k-1})\|_2 > \epsilon/4$ holds with high probability if $\|\nabla f_{\mathcal{S}_k}(\mathbf{x}_{k-1})\|_2 > \epsilon/2$. Similarly, $\|\nabla f(\mathbf{x}_{k-1})\|_2 \leq \epsilon$ holds with high probability if $\|\nabla f_{\mathcal{S}_k}(\mathbf{x}_{k-1})\|_2 \leq \epsilon/2$.

Note that each iteration of the outer loop in Algorithm 2 consists of two cases: (1) if the norm of subsampled gradient $\nabla f_{\mathcal{S}_k}(\mathbf{x}_{k-1})$ is small, then we run one subroutine NCD3-Stochastic, i.e., Algorithm 1; and (2) if the norm of $\nabla f_{\mathcal{S}_k}(\mathbf{x}_{k-1})$ is large, then we run one epoch of SCSG algorithm. This design can reduce the number of negative curvature calculations. There are two major differences between Algorithm 2 and existing algorithms in [34, 5]: (1) the step size of negative curvature descent step in Algorithm 2 is larger; and (2) the minibatch size in each epoch of SCSG in Algorithm 2 can be set to 1 instead of being related to the accuracy parameters $\epsilon$ and $\epsilon_H$, while the minibatch size in each epoch of SCSG in the existing algorithms [34, 5] has to depend on $\epsilon$ and $\epsilon_H$.

Now we present the following theorem which spells out the runtime complexity of Algorithm 2.

**Theorem 5.1.** Let $f(\mathbf{x}) = \mathbb{E}_{\xi \sim \mathcal{D}}[F(\mathbf{x}; \xi)]$. Suppose the third derivative of $f(\mathbf{x})$ is $L_3$-Lipschitz, and each stochastic function $F(\mathbf{x}; \xi)$ is $L_1$-smooth and $L_2$-Hessian Lipschitz continuous. Suppose that the stochastic gradient $\nabla F(\mathbf{x}; \xi)$ satisfies the gradient sub-Gaussian condition with parameter

**Algorithm 2** **F**ast **L**ocal minim**A** finding with third-order **S**moot**H**ness (FLASH-Stochastic)

1: **Input:** $f, \mathbf{x}_0, L_1, L_2, L_3, \delta, \epsilon, \epsilon_H, b, K$
2: Set $B \leftarrow \widetilde{O}(\sigma^2/\epsilon^2), \eta = b^{2/3}/(3L_1 B^{2/3})$
3: **for** $k = 1, 2, ..., K$
4:     uniformly sample a batch $\mathcal{S}_k \sim \mathcal{D}$ with $|\mathcal{S}_k| = B$
5:     $\mathbf{g}_k \leftarrow \nabla f_{\mathcal{S}_k}(\mathbf{x}_{k-1})$
6:     **if** $\|\mathbf{g}_k\|_2 > \epsilon/2$
7:         generate $T_k \sim \text{Geom}(B/(B+b))$
8:         $\mathbf{y}_0^{(k)} \leftarrow \mathbf{x}_{k-1}$
9:         **for** $t = 1, ..., T_k$
10:           randomly pick $\mathcal{I}_t \subset \mathcal{D}$ with $|\mathcal{I}_t| = b$
11:           $\boldsymbol{\nu}_{t-1}^{(k)} \leftarrow \nabla f_{\mathcal{I}_t}(\mathbf{y}_{t-1}^{(k)}) - \nabla f_{\mathcal{I}_t}(\mathbf{y}_0^{(k)}) + \mathbf{g}_k$
12:           $\mathbf{y}_t^{(k)} \leftarrow \mathbf{y}_{t-1}^{(k)} - \eta \boldsymbol{\nu}_{t-1}^{(k)}$
13:         **end for**
14:         $\mathbf{x}_k \leftarrow \mathbf{y}_{T_k}^{(k)}$
15:     **else**
16:         $\mathbf{x}_k \leftarrow$ NCD3-Stochastic$(f, \mathbf{x}_{k-1}, \{L_i\}_{i=1}^3, \delta, \epsilon_H)$
17:         **if** $\mathbf{x}_k = \bot$
18:           **return** $\mathbf{x}_{k-1}$
19: **end for**

$\sigma$. Set batch size $B = \widetilde{O}(\sigma^2/\epsilon^2)$ and $\epsilon_H \gtrsim \epsilon^{2/3}$. If Algorithm 2 adopts online Oja's algorithm to compute the negative curvature, then Algorithm 2 finds an $(\epsilon, \epsilon_H)$-second-order stationary point with probability at least $1/3$ in runtime

$$\widetilde{O}\bigg(\bigg(\frac{L_1 \sigma^{4/3} \Delta_f}{\epsilon^{10/3}} + \frac{L_3 \sigma^2 \Delta_f}{\epsilon^2 \epsilon_H^2}\bigg)\mathbb{T}_g + \bigg(\frac{L_1^2 L_3 \Delta_f}{\epsilon_H^4}\bigg)\mathbb{T}_h\bigg).$$

If Algorithm 2 adopts Neon2$^{\text{online}}$, then it finds an $(\epsilon, \epsilon_H)$-second-order stationary point with probability at least $1/3$ in runtime

$$\widetilde{O}\bigg(\bigg(\frac{L_1 \sigma^{4/3} \Delta_f}{\epsilon^{10/3}} + \frac{L_3 \sigma^2 \Delta_f}{\epsilon^2 \epsilon_H^2} + \frac{L_1^2 L_3 \Delta_f}{\epsilon_H^4}\bigg)\mathbb{T}_g\bigg).$$

**Remark 5.2.** Although the runtime complexity in Theorem 5.1 holds with a constant probability, one can repeatedly run Algorithm 2 for at most $\log(1/\delta)$ times to achieve a high probability result with probability at least $1 - \delta$.

**Remark 5.3.** Theorem 5.1 suggests that the runtime complexity of Algorithm 2 is $\widetilde{O}(\epsilon^{-10/3} + \epsilon^{-2}\epsilon_H^{-2} + \epsilon_H^{-4})$ to find an $(\epsilon, \epsilon_H)$-second-order stationary point. Compared with $\widetilde{O}(\epsilon^{-10/3} + \epsilon^{-2}\epsilon_H^{-3} + \epsilon_H^{-5})$ runtime complexity achieved by the state-of-the-art [5], the runtime complexity of Algorithm 2 is improved upon the state-of-the-art in the second and third terms. If we set $\epsilon_H = \sqrt{\epsilon}$, the runtime of Algorithm 2 is $\widetilde{O}(\epsilon^{-10/3})$ and that of the state-of-the-art stochastic local minima finding algorithms [2, 31, 34, 5] becomes $\widetilde{O}(\epsilon^{-7/2})$, thus Algorithm 2 outperforms the state-of-the-art algorithms by a factor of $\widetilde{O}(\epsilon^{-1/6})$.

**Remark 5.4.** Note that we can set $\epsilon_H$ to a smaller value, i.e., $\epsilon_H = \epsilon^{2/3}$, and the total runtime complexity of Algorithm 2 remains $\widetilde{O}(\epsilon^{-10/3})$. It is also worth noting that the runtime complexity of Algorithm 2 matches that of the state-of-the-art stochastic optimization algorithm (SCSG) [25] which only finds first-order stationary points but does not impose the third-order smoothness assumption.

## 6 Experiments

In this section, we conduct numerical experiment on two nonconvex optimization problems, i.e., matrix sensing and deep autoencoder. All the experiments are carried on Amazon AWS p2.xlarge nodes with NVIDIA GK210 GPUs, and we use Pytorch 0.3.0 to implement all the algorithms.

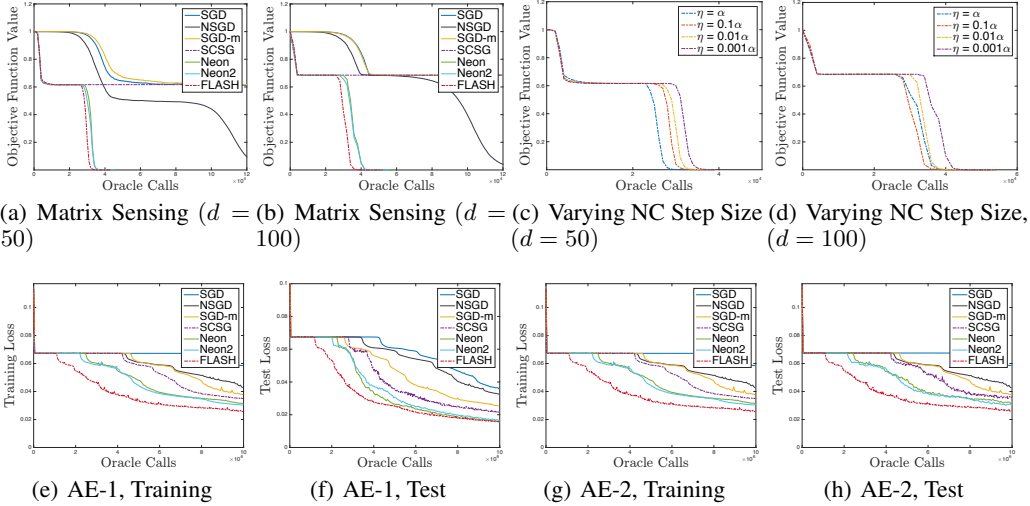

(a) Matrix Sensing ($d =$ 50)  (b) Matrix Sensing ($d =$ 100)  (c) Varying NC Step Size ($d = 50$)  (d) Varying NC Step Size, ($d = 100$)

(e) AE-1, Training  (f) AE-1, Test  (g) AE-2, Training  (h) AE-2, Test

Figure 1: Numerical results for matrix sensing and deep autoencoder. (a)-(b) Convergence of different algorithms for matrix sensing: objective function value versus the number of oracle calls. (c)-(d) Different negative curvature step size comparison of FLASH for matrix sensing. (e)-(h) Convergence of different algorithms for two deep autoencoders: Training loss versus the number of oracle calls and test loss versus the number of oracle calls.

**Matrix Sensing** We consider the symmetric matrix sensing problem, which is defined as:

$$\min_{\mathbf{U} \in \mathbb{R}^{d \times r}} f(\mathbf{U}) = \frac{1}{2m} \sum_{i=1}^{m} \left( \langle \mathbf{A}_i, \mathbf{U}\mathbf{U}^\top \rangle - b_i \right)^2, \tag{6.1}$$

where the matrices $\{\mathbf{A}_i\}_{i=1,\dots,m}$ are known sensing matrices, $b_i = \langle \mathbf{A}_i, \mathbf{M}^* \rangle$ is the $i$-th observation, and $\mathbf{M}^* = \mathbf{U}^*(\mathbf{U}^*)^\top$ is an unknown low-rank matrix, which needs to be recovered. For the data generation, we consider two matrix sensing problems: (1) $d = 50, r = 3$, and (2) $d = 100, r = 3$, then generate $m = 20d$ sensing matrices $\mathbf{A}_1, \dots, \mathbf{A}_m$, where each element of the sensing matrix $\mathbf{A}_i$ follows i.i.d. standard normal distribution, and the unknown low-rank matrix $\mathbf{M}^*$ as $\mathbf{M}^* = \mathbf{U}^*(\mathbf{U}^*)^\top$, where $\mathbf{U}^* \in \mathbb{R}^{d \times r}$ is randomly generated, and thus $b_i = \langle \mathbf{A}_i, \mathbf{M}^* \rangle$. Next we randomly initialize a vector $\mathbf{u}_0 \in \mathbb{R}^d$ satisfying $\|\mathbf{u}_0\|_2 < \lambda_{\max}(\mathbf{M}^*)$ and set the initial input $\mathbf{U}_0$ as $\mathbf{U}_0 = [\mathbf{u}_0, \mathbf{0}, \dots, \mathbf{0}]$.

**Deep Autoencoder** We also perform experiments of training a deep autoencoder on MNIST dataset [19]. The MNIST dataset contains images of handwritten digits, including $60,000$ training examples and $10,000$ test examples. Each image has $28 \times 28$ pixels. We consider two autoencoders: (1) a fully connected encoder with layers of size $(28 \times 28)$-1024-512-256-32 and a symmetric decoder (AE-1) and (2) a fully connected encoder with layers of size $(28 \times 28)$-1024-512-256-128-56-32 and a symmetric decoder (AE-2);. The code layer with 32 units are linear and we use softplus function as the activation function for other layers. We use mean squared error (MSE) as the loss function.

We evaluate our algorithm FLASH-Stochastic (**FLASH** for short) together with the following state-of-the-art stochastic optimization algorithms for nonconvex problems: (1) stochastic gradient descent (**SGD**); (2) SGD with momentum (**SGD-m**); (3) noisy stochastic gradient descent (**NSGD**) [15]; (4) Stochastically Controlled Stochastic Gradient (**SCSG**) [25]; (5) NEgative-curvature-Originated-from-Noise (**Neon**) [34]; (6) NEgative-curvature-Originated-from-Noise 2 (**Neon2**) [5]. A fixed gradient mini-batch size of 100 is used for all the algorithms. We apply Oja's algorithm with a Hessian mini-batch size of 100 to calculate the negative curvature in FLASH. We perform a grid search over step sizes for each method. For the negative curvature step size $\alpha$, we choose $\alpha = O(\epsilon_H/L_2$ for Neon, Neon2 and $\alpha = O(\sqrt{\epsilon_H/L_3})$ for our algorithm FLASH according to the corresponding theories, where $\epsilon_H = 0.001$, and tune the constant parameter in the negative curvature step size by grid search. We report the objective function value versus oracle calls on matrix sensing and training loss versus oracle calls on matrix sensing and deep autoencoder.

The experimental results of the above two nonconvex problems are shown in Figure 1. For the matrix sensing problem, in Figure 1(a)-1(b), we observe that without adding noise or using second-order information, SGD, SGD-m and SCSG are not able to escape from saddle points. Our algorithm and NSGD, Neon, Neon2 can escape from saddle points, and our algorithm converges to the unknown matrix faster than NSGD, Neon, Neon2. As we can see from Figure 1(e)-1(h), for deep autoencoder, compared with SGD, SGD-m, NSGD, SCSG, Neon and Neon2, our algorithm escapes from saddle points faster and converges faster. Our algorithm outperforms Neon and Neon2 on both problems and validates our theoretical analysis that negative curvature step with a larger step size is helpful in stochastic nonconvex optimization problems. We also compare the convergence behavior of our algorithm with different step sizes for negative curvature descent. We first set initial step size $\alpha = 0.2$ (for negative curvature descent) and then decrease the step size by a factor of $0.1$ each time, while the other parameters remain the same. We can see from Figure 1(c) and 1(d) that our algorithm FLASH converges faster with larger step sizes for negative curvature descent, which validates our theories on third-order smoothness can be helpful in the nonconvex stochastic optimization.

## 7 Conclusions

In this paper, we investigated the benefit of third-order smoothness of nonconvex objective functions in stochastic optimization. We illustrated that third-order smoothness can help faster escape saddle points, by proposing a new negative curvature descent algorithms with improved theoretical guarantee. Based on the proposed negative curvature descent algorithm, we further proposed a practical stochastic optimization algorithm with improved run time complexity that finds local minima for stochastic nonconvex optimization problems.

## Acknowledgements

We would like to thank the anonymous reviewers for their helpful comments, and Yu Chen, Xuwang Yin for their helpful discussions on the experiments. This research was sponsored in part by the National Science Foundation IIS-1652539 and BIGDATA IIS-1855099. We also thank AWS for providing cloud computing credits associated with the NSF BIGDATA award. The views and conclusions contained in this paper are those of the authors and should not be interpreted as representing any funding agencies.

## Footnotes

[2]As shown in [9], the second-order accuracy parameter $\epsilon_H$ can be set as $\epsilon^{2/3}$ and the total runtime complexity remains the same, i.e., $\widetilde{O}(\epsilon^{-5/3})$.

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
