[Supplementary Material]

## A   More Discussion on Finding the Negative Curvature

In this section, we present the gradient complexities of the negative curvature finding algorithms used in Section 4. Note that we use ApproxNC-Stochastic in Algorithm 1 to find the negative curvature direction, which is usually done by Oja's algorithm or Neon2$^{\text{online}}$ in stochastic nonconvex optimization problem (1.1). The following lemma characterizes the Hessian-vector product complexity of Oja's algorithm.

**Lemma A.1.** [4] Let $f(\mathbf{x}) = \mathbb{E}_{\xi \sim \mathcal{D}}[F(\mathbf{x}; \xi)]$, where each component function $F(\mathbf{x}; \xi)$ is twice-differentiable and $L_1$-smooth. For any given point $\mathbf{x} \in \mathbb{R}^d$, if $\lambda_{\min}(\nabla^2 f(\mathbf{x})) \leq -\epsilon_H$, then with probability at least $1 - \delta$, Oja's algorithm returns a unit vector $\widehat{\mathbf{v}}$ satisfying

$$\widehat{\mathbf{v}}^\top \nabla^2 f(\mathbf{x}) \widehat{\mathbf{v}} < -\frac{\epsilon_H}{2},$$

with $O\big((L_1^2/\epsilon_H^2) \log^2(d/\delta) \log(1/\delta)\big)$ stochastic Hessian-vector product evaluations.

Next we present the gradient complexity for Neon2$^{\text{online}}$ [5] in the stochastic setting.

**Lemma A.2.** [5] Let $f(\mathbf{x}) = \mathbb{E}_{\xi \sim \mathcal{D}}[F(\mathbf{x}; \xi)]$ where each component function $F(\mathbf{x}; \xi)$ is $L_1$-smooth and $L_2$-Hessian Lipschitz continuous. For any given point $\mathbf{x} \in \mathbb{R}^d$, with probability at least $1 - \delta$, Neon2$^{\text{online}}$ returns $\widehat{\mathbf{v}}$ satisfying one of the following conditions,

- $\widehat{\mathbf{v}} = \perp$, then $\lambda_{\min}(\nabla^2 f(\mathbf{x})) \geq -\epsilon_H$.

- $\widehat{\mathbf{v}} \neq \perp$, then $\widehat{\mathbf{v}}^\top \nabla^2 f(\mathbf{x}) \widehat{\mathbf{v}} \leq -\epsilon_H/2$ with $\|\mathbf{v}\|_2 = 1$.

The total number of stochastic gradient evaluations is $O\big((L_1^2/\epsilon_H^2) \log^2(d/\delta)\big)$.

## B   Revisit of the SCSG Algorithm

In this section, for the purpose of self-containedness, we introduce the nonconvex stochastically controlled stochastic gradient (SCSG) algorithm [25] for general smooth nonconvex optimization problems with finite-sum structure, which is described in Algorithm 3.

---
**Algorithm 3** SCSG $(f, \mathbf{x}_0, T, \eta, B, b, \epsilon)$

---
1: **initialization:** $\widetilde{\mathbf{x}}_0 = \mathbf{x}_0$
2: **for** $k = 1, 2, ..., K$
3:   uniformly sample a batch $\mathcal{S}_k \subset [n]$ with $|\mathcal{S}_k| = B$
4:   $\mathbf{g}_k \leftarrow \nabla f_{\mathcal{S}_k}(\widetilde{\mathbf{x}}_{k-1})$
5:   $\mathbf{x}_0^{(k)} \leftarrow \widetilde{\mathbf{x}}_{k-1}$
6:   generate $T_k \sim \text{Geom}(B/(B+b))$
7:   **for** $t = 1, ..., T_k$
8:     randomly pick $\widetilde{\mathcal{I}}_{t-1} \subset [n]$ with $|\widetilde{\mathcal{I}}_{t-1}| = b$
9:     $\boldsymbol{\nu}_{t-1}^{(k)} \leftarrow \nabla f_{\widetilde{\mathcal{I}}_{t-1}}(\mathbf{x}_{t-1}^{(k)}) - \nabla f_{\widetilde{\mathcal{I}}_{t-1}}(\mathbf{x}_0^{(k)}) + \mathbf{g}_k$
10:     $\mathbf{x}_t^{(k)} \leftarrow \mathbf{x}_{t-1}^{(k)} - \eta \boldsymbol{\nu}_{t-1}^{(k)}$
11:   **end for**
12:   $\widetilde{\mathbf{x}}_k \leftarrow \mathbf{x}_{T_k}^{(k)}$
13: **end for**
14: **output:** Sample $\widetilde{\mathbf{x}}_K^*$ from $\{\widetilde{\mathbf{x}}_k\}_{k=1}^K$ uniformly.

---

The following lemma characterizes the function value gap after one epoch of Algorithm 3, which is a restatement of Theorem 3.1 in [25].

**Lemma B.1.** [25] Let each $f_i$ be $L_1$-smooth. Set $\eta L_1 = \gamma(B/b)^{-2/3}$, $\gamma \leq 1/3$ and $B \geq 8b$. Then at the end of the $k$-th outer loop of Algorithm 3, it holds that

$$\mathbb{E}[\|\nabla f(\widetilde{\mathbf{x}}_k)\|_2^2] \leq \frac{5L_1}{\gamma}\left(\frac{b}{B}\right)^{1/3} \mathbb{E}[f(\mathbf{x}_0^{(k)}) - f(\widetilde{\mathbf{x}}_k)] + \frac{6 \mathbb{1}\{B < n\}}{B}\mathcal{V},$$

where $\mathcal{V}$ is the upper bound on the variance of the stochastic gradient.

Next, we present a general extension of Lemma B.1 to the general stochastic setting in (1.1). In this case, we have $\mathbf{g}_k = \nabla f_{\mathcal{S}_k}(\widetilde{\mathbf{x}}_{k-1}) = 1/B \sum_{i \in \mathcal{S}_k} \nabla F(\widetilde{\mathbf{x}}_{k-1}; \xi_i)$ and $n$ is relatively large, i.e., $n \gg O(1/\epsilon^2)$. Note that by [32] the sub-Gaussian stochastic gradient in Definition 3.7 implies $\mathbb{E}[\|\nabla F(\mathbf{x}; \xi) - \nabla f(\mathbf{x})\|] \leq 2\sigma^2$ and thus we replace $\mathcal{V} = 2\sigma^2$ in Corollary B.1. Then we have the following corollary.

**Corollary B.2.** Let each stochastic function $F(\mathbf{x}; \xi)$ be $L_1$-smooth and suppose that $\nabla F(\mathbf{x}; \xi)$ satisfies the gradient sub-Gaussian condition in Definition 3.7. Suppose that $n \gg O(1/\epsilon^2)$ and $n > B$. Set parameters $b \leq B/8$ and $\eta = b^{2/3}/(3L_1 B^{2/3})$. Then at the end of the $k$-th outer loop of Algorithm 3, it holds that

$$\mathbb{E}[\|\nabla f(\widetilde{\mathbf{x}}_k)\|_2^2] \leq \frac{15 \, b^{1/3} L_1}{B^{1/3}} \mathbb{E}[f(\widetilde{\mathbf{x}}_0^{(k)}) - f(\widetilde{\mathbf{x}}_k)] + \frac{12\sigma^2}{B}.$$

# C   Proofs for Negative Curvature Descent

In this section, we first prove the lemma that characterizes the function value decrease in our negative curvature descent algorithm, i.e., Algorithm 1.

## C.1   Proof of Lemma 4.1

*Proof.* By assumptions, $f(\mathbf{x})$ is $L_3$-Hessian Lipschitz continuous, according to Lemma 1 in [6], for any $\mathbf{x}, \mathbf{y} \in \mathbb{R}^d$, we have

$$f(\mathbf{y}) \leq f(\mathbf{x}) + \langle \nabla f(\mathbf{x}), \mathbf{y} - \mathbf{x} \rangle + \frac{1}{2}(\mathbf{y} - \mathbf{x})^\top \nabla^2 f(\mathbf{x})(\mathbf{y} - \mathbf{x})$$

$$+ \frac{1}{6}\langle \nabla^3 f(\mathbf{x}), (\mathbf{y} - \mathbf{x})^{\otimes 3} \rangle + \frac{L_3}{24}\|\mathbf{y} - \mathbf{x}\|_2^4.$$

Denote the input point $\mathbf{x}$ of Algorithm 1 as $\mathbf{y}_0$. Suppose that $\widehat{\mathbf{v}} \neq \perp$. By Lemmas A.1 and A.2, the ApproxNC-Stochastic algorithm returns a unit vector $\widehat{\mathbf{v}}$ such that

$$\widehat{\mathbf{v}}^\top \nabla^2 f(\mathbf{y}_0)\widehat{\mathbf{v}} \leq -\frac{\epsilon_H}{2} \tag{C.1}$$

holds with probability at least $1 - \delta$ within $\widetilde{O}(L_1^2/\epsilon_H^2)$ evaluations of stochastic Hessian-vector product or stochastic gradient. Define $\mathbf{u} = \mathbf{y}_0 + \alpha\widehat{\mathbf{v}}$ and $\mathbf{w} = \mathbf{y}_0 - \alpha\widehat{\mathbf{v}}$. Then it holds that

$$\sum_{\mathbf{y} \in \{\mathbf{u}, \mathbf{w}\}} \left( \langle \nabla f(\mathbf{y}_0), \mathbf{y} - \mathbf{y}_0 \rangle + \frac{1}{6}\langle \nabla^3 f(\mathbf{y}_0), (\mathbf{y} - \mathbf{y}_0)^{\otimes 3} \rangle \right) = 0.$$

Furthermore, recall that we have $\widehat{\mathbf{y}} = \mathbf{x} + \zeta\alpha\widehat{\mathbf{v}}$ in Algorithm 1 where $\zeta$ is a Rademacher random variable and thus we have $\mathbb{P}(\zeta = 1) = \mathbb{P}(\widehat{\mathbf{y}} = \mathbf{u}) = 1/2$ and $\mathbb{P}(\zeta = -1) = \mathbb{P}(\widehat{\mathbf{y}} = \mathbf{w}) = 1/2$, which immediately implies

$$\begin{aligned}
\mathbb{E}_\zeta[f(\widehat{\mathbf{y}}) - f(\mathbf{y}_0)] &\leq \frac{1}{2}(\widehat{\mathbf{y}} - \mathbf{y}_0)^\top \nabla^2 f(\mathbf{y}_0)(\widehat{\mathbf{y}} - \mathbf{y}_0) + \frac{L_3}{24}\|\widehat{\mathbf{y}} - \mathbf{y}_0\|_2^4 \\
&\leq \frac{\alpha^2}{2}\widehat{\mathbf{v}}^\top \nabla^2 f(\mathbf{y}_0)\widehat{\mathbf{v}} + \frac{L_3\alpha^4}{24}\|\widehat{\mathbf{v}}\|_2^4 \\
&\leq -\frac{\alpha^2}{2}\frac{\epsilon_H}{2} + \frac{L_3\alpha^4}{24}\|\widehat{\mathbf{v}}\|_2^4 \\
&= -\frac{3\epsilon_H^2}{8L_3} \tag{C.2}
\end{aligned}$$

holds with probability at least $1 - \delta$, where $\mathbb{E}_\zeta$ denotes the expectation over $\zeta$, the third inequality follows from (C.1) and in the last equality we used the fact that $\alpha = \sqrt{3\epsilon_H/L_3}$. On the other hand, by $L_2$-smoothness of $f$ (as each stochastic function $F(\mathbf{x}; \xi)$ is $L_2$-Hessian Lipschitz continuous), we can derive that

$$\begin{aligned}
\mathbb{E}_\zeta[f(\widehat{\mathbf{y}}) - f(\mathbf{y}_0)] &\leq \frac{\alpha^2}{2}\widehat{\mathbf{v}}^\top \nabla^2 f(\mathbf{y}_0)\widehat{\mathbf{v}} + \frac{L_3\alpha^4}{24}\|\widehat{\mathbf{v}}\|_2^4 \\
&\leq \frac{\alpha^2 L_2}{2} + \frac{L_3\alpha^4}{24}\|\widehat{\mathbf{v}}\|_2^4 \\
&= \frac{3\epsilon_H(\epsilon_H + 4L_2)}{8L_3}. \tag{C.3}
\end{aligned}$$

Combining (C.2) and (C.3) yields

$$\mathbb{E}[f(\widehat{\mathbf{y}}) - f(\mathbf{y}_0)] \leq -\frac{3(1-\delta)\epsilon_H^2}{8L_3} + \frac{3\delta\epsilon_H(\epsilon_H + 4L_2)}{8L_3}$$
$$\leq -\frac{3(1-\delta)\epsilon_H^2}{16L_3},$$

where the second inequality holds if $\delta \leq \epsilon_H/(3\epsilon_H + 8L_2)$. Furthermore, plugging $\delta < 1/3$ into the above inequality we obtain $\mathbb{E}[f(\widehat{\mathbf{y}}) - f(\mathbf{y}_0)] \leq -\epsilon_H^2/(8L_3)$. □

# D  Proofs for Runtime Complexity of Algorithms

In this section, we prove the main theorem for our stochastic local minima finding algorithm.

## D.1  Proof of Theorem 5.1

Before we prove theoretical results for the stochastic setting, we lay down the following useful lemma which states the concentration bound for sub-Gaussian random vectors.

**Lemma D.1.** [17] Suppose the stochastic gradient $\nabla F(\mathbf{x}; \xi)$ is sub-Gaussian with parameter $\sigma$. Let $\nabla f_{\mathcal{S}}(\mathbf{x}) = 1/|\mathcal{S}| \sum_{i \in \mathcal{S}} \nabla F(\mathbf{x}; \xi_i)$ be a subsampled gradient of $f$. If the sample size $|\mathcal{S}| = 2\sigma^2/\epsilon^2(1 + \sqrt{\log(1/\delta)})^2$, then with probability $1 - \delta$,

$$\|\nabla f_{\mathcal{S}}(\mathbf{x}) - \nabla f(\mathbf{x})\|_2 \leq \epsilon$$

holds for any $\mathbf{x} \in \mathbb{R}^d$.

*Proof of Theorem 5.1.* We first calculate the outer loop iteration complexity of Algorithm 2. Let $\mathcal{I} = \{1, 2, \ldots, K\}$ be the index of iteration. We use $\mathcal{I}_1$ and $\mathcal{I}_2$ to denote the index set of iterates that are output by the NCD3-Stochastic stage and SCSG stage of Algorithm 2 respectively. It holds that $K = |\mathcal{I}| = |\mathcal{I}_1| + |\mathcal{I}_2|$. We will calculate $|\mathcal{I}_1|$ and $|\mathcal{I}_2|$ in sequence.

**Computing $|\mathcal{I}_1|$**: note that $|\mathcal{I}_1|$ is the number of iterations that Algorithm 2 calls NCD3-Stochastic to find the negative curvature. Recall the result in Lemma 4.1, for $k \in \mathcal{I}_1$, one execution of the NCD3-Stochastic stage can reduce the function value up to

$$\mathbb{E}[f(\mathbf{x}_{k-1}) - f(\mathbf{x}_k)] \geq \frac{\epsilon_H^2}{8L_3}. \tag{D.1}$$

To get the upper bound of $|\mathcal{I}_1|$, we also need to consider iterates output by One-epoch SCSG. By Lemma B.2 it holds that

$$\mathbb{E}[\|\nabla f(\mathbf{x}_k)\|_2^2] \leq \frac{C_1 L_1}{B^{1/3}} \mathbb{E}[f(\mathbf{x}_{k-1}) - f(\mathbf{x}_k)] + \frac{C_2 \sigma^2}{B}, \quad \text{for } k \in \mathcal{I}_2, \tag{D.2}$$

where $C_1 = 15b^{1/3}$, $C_2 = 12$ are absolute constants, and we assume $b \leq B/8$.
As for $k \in \mathcal{I}_2$, we further decompose $\mathcal{I}_2$ as $\mathcal{I}_2 = \mathcal{I}_2^1 \cup \mathcal{I}_2^2$, where $\mathcal{I}_2^1 = \{k \in \mathcal{I}_2 \mid \|\mathbf{g}_k\|_2 > \epsilon/2\}$ and $\mathcal{I}_2^2 = \{k \in \mathcal{I}_2 \mid \|\mathbf{g}_k\|_2 \leq \epsilon/2\}$. It is easy to see that $\mathcal{I}_2^1 \cap \mathcal{I}_2^2 = \varnothing$ and $|\mathcal{I}_2| = |\mathcal{I}_2^1| + |\mathcal{I}_2^2|$. In addition, according to the concentration result on $\mathbf{g}_k$ and $\nabla f(\mathbf{x}_k)$ in Lemma D.1, if the sample size $B$ satisfies $B = O(\sigma^2/\epsilon^2 \log(1/\delta_0))$, then for any $k \in \mathcal{I}_2^1$, $\|\nabla f(\mathbf{x}_k)\|_2 > \epsilon/4$ holds with probability at least $1 - \delta_0$. For any $k \in \mathcal{I}_2^2$, $\|\nabla f(\mathbf{x}_k)\|_2 \leq \epsilon$ holds with probability at least $1 - \delta_0$. According to (D.2), we can derive that for any $k \in \mathcal{I}_2^1$,

$$\mathbb{E}[f(\mathbf{x}_{k-1}) - f(\mathbf{x}_k)] \geq \frac{B^{1/3}}{C_1 L_1} \mathbb{E}[\|\nabla f(\mathbf{x}_k)\|_2^2] - \frac{C_2 \sigma^2}{C_1 L_1 B^{2/3}}, \quad \text{for } k \in \mathcal{I}_2^1. \tag{D.3}$$

As for $|\mathcal{I}_2^2|$, because for any $k \in \mathcal{I}_2^2$, $\|\mathbf{g}_k\|_2 \leq \epsilon/2$, which will lead the algorithm to execute one step of NCD3-Stochastic stage in the next iteration, i.e., $k$-th iteration. Thus it immediately implies that $|\mathcal{I}_2^2| \leq |\mathcal{I}_1|$, and according to (D.2), we can also derive that

$$\mathbb{E}[f(\mathbf{x}_{k-1}) - f(\mathbf{x}_k)] \geq -\frac{C_2 \sigma^2}{C_1 L_1 B^{2/3}}, \quad \text{for } k \in \mathcal{I}_2^2. \tag{D.4}$$

Summing up (D.1) over $k \in \mathcal{I}_1$, (D.3) over $k \in \mathcal{I}_2^1$, (D.4) over $k \in \mathcal{I}_2^2$ and combining the results yields

$$\sum_{k \in \mathcal{I}} \mathbb{E}[f(\mathbf{x}_{k-1}) - f(\mathbf{x}_k)] \geq \sum_{k \in \mathcal{I}_1} \frac{\epsilon_H^2}{8L_3} + \frac{B^{1/3}}{C_1 L_1} \sum_{k \in \mathcal{I}_2^1} \mathbb{E}[\|\nabla f(\mathbf{x}_k)\|_2^2]$$
$$- \sum_{k \in \mathcal{I}_2^1} \frac{C_2 \sigma^2}{C_1 L_1 B^{2/3}} - \sum_{k \in \mathcal{I}_2^2} \frac{C_2 \sigma^2}{C_1 L_1 B^{2/3}},$$

which immediately implies

$$\frac{|\mathcal{I}_1| \epsilon_H^2}{8L_3} + \frac{B^{1/3}}{C_1 L_1} \sum_{k \in \mathcal{I}_2^1} \mathbb{E}[\|\nabla f(\mathbf{x}_k)\|_2^2] \leq \Delta_f + \sum_{k \in \mathcal{I}_2^1} \frac{C_2 \sigma^2}{C_1 L_1 B^{2/3}} + \sum_{k \in \mathcal{I}_2^2} \frac{C_2 \sigma^2}{C_1 L_1 B^{2/3}}$$
$$\leq \Delta_f + \frac{|\mathcal{I}_2^1| C_2 \sigma^2}{C_1 L_1 B^{2/3}} + \frac{|\mathcal{I}_1| C_2 \sigma^2}{C_1 L_1 B^{2/3}},$$

where the first inequality uses the fact that $\Delta_f = f(\mathbf{x}_0) - \inf_{\mathbf{x}} f(\mathbf{x})$ and the second inequality is due to $|\mathcal{I}_2^2| \leq |\mathcal{I}_1|$. Applying Markov inequality to the left-hand side of the first inequality above we have that

$$\frac{|\mathcal{I}_1| \epsilon_H^2}{8L_3} + \frac{B^{1/3}}{C_1 L_1} \sum_{k \in \mathcal{I}_2^1} \|\nabla f(\mathbf{x}_k)\|_2^2 \leq 3\mathbb{E}\left[\frac{|\mathcal{I}_1| \epsilon_H^2}{8L_3} + \frac{B^{1/3}}{C_1 L_1} \sum_{k \in \mathcal{I}_2^1} \|\nabla f(\mathbf{x}_k)\|_2^2\right]$$
$$\leq 3\Delta_f + \frac{3|\mathcal{I}_2^1| C_2 \sigma^2}{C_1 L_1 B^{2/3}} + \frac{3|\mathcal{I}_1| C_2 \sigma^2}{C_1 L_1 B^{2/3}}$$

holds with probability at least $1 - 1/3 = 2/3$. Note that $\|\nabla f(\mathbf{x}_k)\|_2 \geq \epsilon/4$ with probability at least $1 - \delta_0$. We conclude that by union bound we have that

$$\frac{|\mathcal{I}_1| \epsilon_H^2}{8L_3} + \frac{|\mathcal{I}_2^1| B^{1/3} \epsilon^2}{16 C_1 L_1} \leq 3\Delta_f + \frac{3|\mathcal{I}_2^1| C_2 \sigma^2}{C_1 L_1 B^{2/3}} + \frac{3|\mathcal{I}_1| C_2 \sigma^2}{C_1 L_1 B^{2/3}}$$

holds with probability at least $2/3(1 - \delta_0)^{|\mathcal{I}_2^1|}$. We can set $B$ such that

$$\frac{3 C_2 \sigma^2}{C_1 L_1 B^{2/3}} \leq \frac{\epsilon_H^2}{16 L_3},$$

which implies

$$B \geq \left(\frac{48 C_2 L_3 \sigma^2}{C_1 L_1}\right)^{3/2} \frac{1}{\epsilon_H^3}. \tag{D.5}$$

Combining the above two inequalities yields

$$\frac{|\mathcal{I}_1| \epsilon_H^2}{16 L_3} + \frac{|\mathcal{I}_2^1| B^{1/3} \epsilon^2}{16 C_1 L_1} \leq 3\Delta_f + \frac{3|\mathcal{I}_2^1| C_2 \sigma^2}{C_1 L_1 B^{2/3}} \tag{D.6}$$

holds with probability at least $2/3(1 - \delta_0)^{|\mathcal{I}_2^1|}$. Therefore, it holds with probability at least $2/3(1 - \delta_0)^{|\mathcal{I}_2^1|}$ that

$$|\mathcal{I}_1| \leq \frac{48 L_3 \Delta_f}{\epsilon_H^2} + \frac{48 C_2 L_3 \sigma^2}{C_1 L_1 B^{2/3} \epsilon_H^2} |\mathcal{I}_2^1| = O\left(\frac{L_3 \Delta_f}{\epsilon_H^2}\right) + \widetilde{O}\left(\frac{L_3 \sigma^2}{L_1 B^{2/3} \epsilon_H^2}\right) |\mathcal{I}_2^1|. \tag{D.7}$$

As we can see from the above inequality, the upper bound of $|\mathcal{I}_1|$ is related to the upper bound of $|\mathcal{I}_2^1|$. We will derive the upper bound on $|\mathcal{I}_2^1|$ later.

**Computing $|\mathcal{I}_2|$:** we have shown that $|\mathcal{I}_2^2| \leq |\mathcal{I}_1|$. Thus we only need to compute the cardinality of subset $\mathcal{I}_2^1 \subset \mathcal{I}_2$, where $\|\mathbf{g}_k\|_2 > \epsilon/2$ for any $k \in \mathcal{I}_2^1$. By Lemma D.1 we can derive that with probability at least $1 - \delta_0$, it holds that $\|\nabla f(\mathbf{x}_k)\|_2 > \epsilon/4$. According to (D.6), we have

$$\frac{|\mathcal{I}_2^1| B^{1/3} \epsilon^2}{16 C_1 L_1} \leq \frac{|\mathcal{I}_1| \epsilon_H^2}{16 L_3} + \frac{|\mathcal{I}_2^1| B^{1/3} \epsilon^2}{16 C_1 L_1} \leq 3\Delta_f + \frac{3|\mathcal{I}_2^1| C_2 \sigma^2}{C_1 L_1 B^{2/3}} \tag{D.8}$$

holds with probability at least $2/3(1 - \delta_0)^{|\mathcal{I}_2^1|}$. Further ensure that $B$ satisfies

$$\frac{3C_2\sigma^2}{C_1 L_1 B^{2/3}} \leq \frac{B^{1/3}\epsilon^2}{32 C_1 L_1},$$

which implies

$$B \geq \frac{96 C_2 \sigma^2}{\epsilon^2}. \tag{D.9}$$

Finally we get the upper bound of $|\mathcal{I}_2^1|$,

$$|\mathcal{I}_2^1| \leq \frac{96 C_1 L_1 \Delta_f}{B^{1/3}\epsilon^2} = \widetilde{O}\left(\frac{L_1 \Delta_f}{\sigma^{2/3}\epsilon^{4/3}}\right), \tag{D.10}$$

where in the equation we use the fact in (D.5) and (D.9), and the condition that $B = \widetilde{O}(\sigma^2/\epsilon^2)$ and $\epsilon_H \gtrsim \epsilon^{2/3}$, which makes the large batch size $B$ also satisfies the condition in (D.5).

More specifically, the starting point to upper bound $|\mathcal{I}_2^1|$ is equation (D.8). We choose sufficient large $B$ (as suggested in equation (D.9)) to ensure the second term in R.H.S of (D.8) is less than half of the L.H.S. of (D.8). Therefore, we can get the upper bound of $|\mathcal{I}_2^1|$ in (D.10).

We then plug the upper bound of $|\mathcal{I}_2^1|$ into (D.7) to obtain the upper bound of $\mathcal{I}_1$. Note that $B = \widetilde{O}(\sigma^2/\epsilon^2)$. Then we have

$$
\begin{aligned}
|\mathcal{I}_1| &\leq \frac{48 L_3 \Delta_f}{\epsilon_H^2} + \frac{48 C_2 L_3 \sigma^2}{C_1 L_1 B^{2/3}\epsilon_H^2}|\mathcal{I}_2^1| \\
&= \widetilde{O}\left(\frac{L_3 \Delta_f}{\epsilon_H^2}\right) + \widetilde{O}\left(\frac{L_3 \sigma^{2/3}\epsilon^{4/3}}{L_1 \epsilon_H^2}\right) \cdot \widetilde{O}\left(\frac{L_1 \Delta_f}{\sigma^{2/3}\epsilon^{4/3}}\right) \\
&= \widetilde{O}\left(\frac{L_3 \Delta_f}{\epsilon_H^2}\right)
\end{aligned}
$$

holds with probability at least $2/3(1 - \delta_0)^{|\mathcal{I}_2^1|}$, where we take $B = \widetilde{O}(\sigma^2/\epsilon^2)$ in the first equality. Choosing sufficient small $\delta_0$ such that $(1 - \delta_0)^{|\mathcal{I}_2^1|} > 1/2$, the upper bound of $\mathcal{I}_1$ and $\mathcal{I}_2^1$ holds with probability at least $1/3$.

**Computing Runtime**: By Lemma 4.1 we know that each call of the NCD3-Stochastic algorithm takes $\widetilde{O}((L_1^2/\epsilon_H^2)\mathbb{T}_h)$ runtime if Oja's algorithm is used and $\widetilde{O}((L_1^2/\epsilon_H^2)\mathbb{T}_g)$ runtime if Neon2$^{\text{online}}$ is used. On the other hand, Corollary B.2 shows that the complexity of one epoch of SCSG algorithm is $\widetilde{O}(\sigma^2/\epsilon^2)$ which implies that the run time of one epoch of SCSG algorithm is $\widetilde{O}((\sigma^2/\epsilon^2)\mathbb{T}_g)$. Therefore, we can compute the total time complexity of Algorithm 2 with online Oja's algorithm as follows

$$
\begin{aligned}
&|\mathcal{I}_1| \cdot \widetilde{O}\left(\frac{L_1^2}{\epsilon_H^2}\mathbb{T}_h\right) + |\mathcal{I}_2| \cdot \widetilde{O}\left(\frac{\sigma^2}{\epsilon^2}\mathbb{T}_g\right) \\
&= |\mathcal{I}_1| \cdot \widetilde{O}\left(\frac{L_1^2}{\epsilon_H^2}\mathbb{T}_h\right) + (|\mathcal{I}_2^1| + |\mathcal{I}_2^2|) \cdot \widetilde{O}\left(\frac{\sigma^2}{\epsilon^2}\mathbb{T}_g\right) \\
&\leq |\mathcal{I}_1| \cdot \widetilde{O}\left(\frac{L_1^2}{\epsilon_H^2}\mathbb{T}_h\right) + (|\mathcal{I}_2^1| + |\mathcal{I}_1|) \cdot \widetilde{O}\left(\frac{\sigma^2}{\epsilon^2}\mathbb{T}_g\right).
\end{aligned}
$$

Plugging the upper bounds of $|\mathcal{I}_1|$ and $|\mathcal{I}_2^1|$ into the above equation yields the following runtime complexity of Algorithm 2 with online Oja's algorithm

$$
\begin{aligned}
&\widetilde{O}\left(\frac{L_3 \Delta_f}{\epsilon_H^2}\right) \cdot \widetilde{O}\left(\frac{L_1^2}{\epsilon_H^2}\mathbb{T}_h\right) + \widetilde{O}\left(\frac{L_1 \Delta_f}{\sigma^{2/3}\epsilon^{4/3}} + \frac{L_3 \Delta_f}{\epsilon_H^2}\right) \cdot \widetilde{O}\left(\frac{\sigma^2}{\epsilon^2}\mathbb{T}_g\right) \\
&= \widetilde{O}\left(\left(\frac{L_1 \sigma^{4/3}\Delta_f}{\epsilon^{10/3}} + \frac{L_3 \sigma^2 \Delta_f}{\epsilon^2 \epsilon_H^2}\right)\mathbb{T}_g + \left(\frac{L_1^2 L_3 \Delta_f}{\epsilon_H^4}\right)\mathbb{T}_h\right),
\end{aligned}
$$

and the runtime complexity of Algorithm 2 with Neon2$^{\text{online}}$ is

$$\widetilde{O}\left(\left(\frac{L_1 \sigma^{4/3}\Delta_f}{\epsilon^{10/3}} + \frac{L_3 \sigma^2 \Delta_f}{\epsilon^2 \epsilon_H^2} + \frac{L_1^2 L_3 \Delta_f}{\epsilon_H^4}\right)\mathbb{T}_g\right),$$

which concludes our proof. $\qquad\square$