[Reviews · NeurIPS 2018]

Reviewer 1



This submission is concerned with unconstrained nonconvex stochastic optimization problems, in a setting in which the function to optimize is only available through stochastic estimates. Obtaining points satisfying second-order necessary optimality conditions has been a recent topic of interest at NIPS, as such points can be as good as global minima on several problems arising from machine learning. The authors present new complexity results that improve over the existing complexity bounds for finding an approximate local minimum, identified as a point for which the gradient norm is less than a threshold $\epsilon$ and the minimum Hessian eigenvalue is at least $-\sqrt{\epsilon}$. By assuming that the objective function possesses a Lipschitz continous third-order derivative, the authors are able to guarantee a larger decrease for steps of negative curvature type: this argument is the key for obtaining lower terms in the final complexity bounds and, as a result, lower dependency on the tolerance $\epsilon$ compared to other techniques ($\epsilon^{-10/3}$ versus $\epsilon^{-7/2}$ in previous works). The authors conduct a thorough review of the related literature, and discuss the main differences between existing algorithms and theirs in Section 2. This literature review appears exhaustive, and identifies key differences between this work and the cited ones. Section 4 presents the source for the improved results, which lies in the use of the Lipschitz constant of the third-order derivative to scale a negative curvature step. This result is, to the authors' own admission, inspired by reference [9] of the submission, and it even seems that the authors could have used some results from [9, Section 4.1] off-the-shelf'' in their analysis, as they have done so for several other lemmas in appendix. I acknowledge that there are differences between their (stochastic) context and that of [9], however I believe the connection with [9] could be better emphasized. In particular, the formula used here for the step size $\eta = \sqrt{3\epsilon_h/L_3}$ already appeared in [9], along with the idea of considering opposite directions for descent. The introduction of a Rademacher variable to cope with stochasticity is an interesting feature, which I would have liked to see discussed more in detail. Section 5 combines the new negative curvature descent routine with the Stochastically Controlled Stochastic Gradient (SCSG) algorithm. The combination of this first-order method with a negative curvature subroutine cannot be viewed as a fully original contribution, since it already appeared in references [5,34], as described by the authors in Section 2. The main complexity bound is presented in this section, along with several remarks comparing the results with those achieved by similar techniques, and the improvement in terms of dependencies with respect to $\epsilon$ is discussed. The use of the new negative curvature descent procedure clearly improves this dependency, yet the knowledge of the third-order Lipschitz constant is necessary to yield this improvement. Section 6 provides numerical experiments on two nonconvex problems for the proposed algorithm, called FLASH, compared to variants of Stochastic Gradient Descent (SGD) and SCSG. The figures indicate that FLASH has a better behavior on those problems. Nevertheless, it seems that none of the other schemes used for comparison achieves a second-order guarantee similar to that of FLASH (the noisy stochastic gradient method provides guarantees of converging close to a local minimum, but under a different set of assumptions). Moreover, it seems that none of the other methods relies on negative curvature steps, Hessian-vector or stochastic Hessian-vector products. Without questioning the interest of the comparison (which advocates for the use of negative curvature directions), one can wonder why the authors did not compare with methods that were augmented with negative curvature subroutines. In particular, given that the behavior of SCSG is already good compared to the SGD-based algorithms on the matrix sensing problem, a comparison with other SCSG-based techniques such as those in [5,34] would have been interesting. If FLASH was shown to outperform those methods, it could be an argument in favor of large steps sizes for negative curvature descent. In summary, I believe the ideas and results of the submission are correct, clearly motivated and described throughout the main body of the submission. As far I could check, the results in the appendices are correct. The originality of the submission is difficult to assess, in that it gather ideas and results for several other works. The main contribution is the use of a new negative curvature subroutine for stochastic optimization, and its analysis is new: however, it is still close to the original idea in the deterministic setting presented in [9]. The hybridization of a negative curvature routine with the SCSG framework is not new, even if the use of this particular routine is. It is unclear whether this submission will have a large impact on the community, especially without comparison with other SCSG-based frameworks. As a result, I will give this submission an overall score of 5. Additional comments: a) Line 280: step size $\alpha$ (What is $\alpha$)" is most likely a typo. I would indeed be interested in knowing the chosen initial value for $\alpha$, as it does not seem that information about the Lipschitz constants was used. b) Lines 404-405: A more precise reference to [32] would be appreciated, to confirm that one indeed obtains $2\sigma^2$ in the variance formula. I was also expecting the norm to be squared. More details would be appreciated. c) Line 457: Applying Markov's inequality [...]" Given that the expectation on the previous equation line 455 does not encompasses the entire left-hand side, the application of Markov's inequality can be unclear at first reading (in particular, one might get confused by the presence of multiple expectations in the sum). I would suggest that the authors add a left-hand side to the last equation on line 455, indicating the expected value on which Markov's inequality will be applied. d) Lines 472-473: The way the upper bound on $|\mathcal{I}^1_2|$ is derived could in my opinion be better described. In particular, it might be good to indicate which equation involving $|\mathcal{I}^1_2|$ is used as a starting point, to which the various bounds on $B$ are then applied. Edit following author feedback: I thank the authors for the efforts they made in addressing my comments during the feedback phase. Their additional numerical experiments are informative, but in my opinion they are not convincing enough to change my review. In particular, I would have liked to be certain that the variants of NEON and NEON2 that are run correspond to a combination of these routines with SCSG, which is not clearly indicated by the authors. Still, assuming that this is the case, these additional experiments show that the performance of FLASH is close to that of NEON/NEON 2 on the MS problem, and it is not clear if the better performance of FLASH is due to longer negative curvature steps. I would have expected the authors to provide evidence that the NEON and NEON2 methods are indeed taking smaller steps in negative curvature directions, which is to be expected from the theoretical analysis. I thus believe that the authors could have made a stronger case for their method by actually showing that longer negative curvature steps were taken, which is not straightforward from their experiments. Since on the theoretical side, my concerns have not been alleviated by the author feedback nor the other reviews, I maintain my original assessment, and a score of 5 for this submission.

Reviewer 2



This paper proposes a novel method to find local minima of non-convex problems. In particular, the proposed method exploits the third-order smoothness to escape saddle points more efficiently. As shown in this paper, the third-order smoothness can help to choose a larger step size along the negative curvature descent direction than existing methods. Overall, the idea is novel and it is well structured. What I am concerned about is as follows: 1.Since this method utilizes the negative curvature direction, it is better to compare it with the corresponding counterparts, such as the Neon algorithm, in the experiment section. 2.Please carefully refine the writing, there are some typos. e.g. Line 288: find->finds

Reviewer 3



This study considers the role of third-order smoothness of nonconvex objective functions in the context of stochastic stochastic optimization. The results show how this can help in escaping saddlepoints. The core result consist in a novel descent algorithm based on negative curvature estimates. The algorithm has decrement guarantees which improve state of the art. The author also design an efficient practical algorithm for escaping saddles. A solid paper which I would accept.